# The Odbert Psalter (Boulogne-sur-Mer, BM, ms. 20); or, the Image as a Medium for Contemplative Practice

Blanche Lagrange 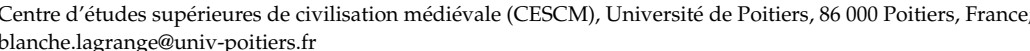

Centre d'études supérieures de civilisation médiévale (CESCM), Université de Poitiers, 86 000 Poitiers, France; blanche.lagrange@univ-poitiers.fr

**Abstract:** The monastic reforms of the 10th century greatly increased the role of the psalter, a biblical book that became the main tool of a monk in personal and collective prayer. The Odbert Psalter, produced in Saint-Bertin around 999, opens with a scene of Pentecost in which we see Christ represented as a king who is static and in a space distinct from the apostles, exhibiting an attitude of meditation. This is not a narrative image: this scene is an indication for the reader of the Psalms. If he follows the example of the apostles, he will arrive at the vision of God, which can only be attained through continuous meditation on the Psalms as it was defined by the reforms. This image serves as a medium for the act of contemplation itself: according to the three modes of vision defined by Saint Augustine, the image of Christ constitutes a pathway from corporeal vision to intellectual vision. By constituting the support of divine contemplation, the psalter and its images are set up here as perfect mediators of the power of the intellect.

**Keywords:** Benedictine rule; reform; psalter; illumination of manuscript; Saint Augustine; intellectual vision

## 1. Introduction

In "contemplative" monastic communities, such as the Benedictines, the search for the "vision" of God is based on the spiritual practices of prayer, meditation and liturgy which punctuate the lives of the monks. At the end of the 10th century, books of prayers and liturgy saw an increase in the images and figurative representations associated with texts. We can suppose that these images served a function in the liturgy similar to prayer, yet they also seem to play a role in the contemplation of God. The Odbert Psalter (Boulogne-sur-Mer, Bibliothèque Municipale, ms. 20) provides an especially good opportunity to examine the dynamics of contemplative acts at the end of the 10th century and the role of the images in it.

The psalter contains a Pentecost scene which is placed within the initial B of the text of the first psalm. The representation of this episode may be surprising: Pentecost, which generally appears at the end of the Christological cycles, here opens the text of the Psalms. It is depicted in an unusual way and not in conformity with the biblical text. What is most surprising is the importance given to the representation of Christ, which is usually absent from this scene. This scene and its location have disconcerted preceding scholars (Leroquais 1941; Kahsnitz 1988, 2004) who have concluded that this arrangement is random without considering whether it could be related to the psalter's use.

How might we explain the singular representation of Pentecost and its location in the psalter? Do the practices surrounding the images make it possible to answer these questions? For this, we must consider these practices, as well as the fundamental role of the psalter in the monastic spirituality of the 10th century. The image of Pentecost will also be analyzed with the help of the systems of vision determined by Saint Augustine. Finally, this essay attempts to prove that the originality of the Pentecost scene can be explained by the preponderant place of the image in contemplative practices. This essay also states that

contemplation is enabled not only by the image but also by its association with the text of Psalms.

## 2. Medieval Practices around the Image

The Odbert Psalter was produced at the Benedictine abbey of Saint-Bertin located in Saint-Omer (Pas-de-Calais, France) at the very end of the 10th century. The *Annals of Saint-Bertin*, dating from the 16th century (Saint-Omer, Bibliothèque d'Agglomération, ms. 747), dates the Odbert Psalter to be precisely from 999. At that time, the scriptorium of the Flemish abbey experienced, under the leadership of Abbot Odbert (986–1007), an intense period of manuscript production. The psalmic text is enriched with figurative scenes, most of which are from the New Testament and depict the life of Christ in 24 historiated initials. These images have been the subject of some research, notably by Victor Leroquais and Rainer Kahsnitz, who tried to identify each scene but did not interpret them. They pointed out the abundance of Christological scenes in the psalter; this abundance makes clear that the images were considered important to the use of the psalter as a religious object.

Indeed, during the Middle Ages, the image was thought to conjoin the material and the spiritual. As Jérôme Baschet states: "to take on the image is necessarily to agree to represent corporeally what, however, must be understood spiritually" ("assumer l'image, c'est nécessairement accepter de representer corporellement ce qui, pourtant, doit s'entendre spirituellement") (Baschet 2016, p. 74). The image represents God or the saints according to the principle of resemblance between the representation and the subject or prototype. This representation "makes present" the prototype: the image shows the active presence of God or the saint, who is represented due to their location in the image. Thus, the image is therefore a transit from the divine to the Earth, a material vehicle for divine powers that ensures the effectiveness of the intercession of the prototype.

Referring to the time during which the Odbert Psalter was produced, Barbara Baert evokes a "wave of *iconophilia*" specific to the 10th century (Baert 2014, p. 86; 2016, p. 525), but this has to be qualified. Instead of stating that for this time period, we can observe a "new theoretical position of images" as Baert does (Baert 2014, p. 86; 2016, p. 525), we can emphasize that the practice surrounding images changed at that time. In his study of images of saints and miracles, Jean-Marie Sansterre argued that by the 11th century, images were venerated in their own right (Sansterre 2006). This worship was not idolatrous: veneration was ultimately directed toward the prototype (God or the saint represented). But according to the logic of the spiritualization of the corporeal proposed by the Church, as Baschet notes (Baschet 2016, p. 73), images did not solely convey the sacrality of the thing they represented; rather, they possessed that sacrality in themselves, and they could therefore serve as vehicles of transport from the material to the spiritual.

While Eastern practices surrounding images suggest that the transitus depends solely on the sacred power of the image, Western practices suggest that it also depends on the Christian (Sansterre and Schmitt 1999). The image stimulates the religious feelings of the Christian, who must then try to find God beyond his image through prayer and meditation and must concentrate his mental and emotional capacities on the subject represented in order to project himself beyond the sensible world. It is a practice in which the Christian must implement a spiritual gaze in order to facilitate a passage toward the divine from the material object. Thanks to the character of imitation between the represented subject and the meditated object, the image therefore serves, as we will see later, as a springboard from projective models to devotional meditation practices. Such interpretations are particularly suited to the image discussed in this article.

## 3. The Odbert Psalter and Its Importance in the Bertinian Monastic Community

The function of this understanding of the images in the Odbert Psalter depends on the changing role of the Psalter itself in the life of the monastic community. The creation of the Odbert Psalter responded to a need for piety and penance related to the Benedictine reforms of the second half of the 10th century, which accorded this book major importance



in liturgy and private prayer (Gretsch 1999, p. 6). The manuscript contains the 150 Psalms and the canticles of the Old Testament, preceded by prologues and followed by hymns. The text of the Psalms is surrounded, in the margins, by a commentary inspired by St. Augustine's Psalms commentary (Augustine of Hippo 2007). It takes its name from the abbot of the Saint-Bertin monastery, Odbert (abbot from around 986 to 1007), who was also the head of the Bertinian scriptorium and an illuminator. The dedication poem at the beginning of the manuscript (folio 1 verso) states: "Me compsit Heriveus, et Odbertus decoravit, excerpsit Dodolinus" ("Heriveus arranged me, Odbert decorated me, Dodolinus made the selection"). Here, "decoravit", from the Latin word "decor", designates what is beautiful, but it also manifests the dignity of this object, of its function, of its status. The term "decoravit", therefore, matches beauty with the essence of the thing evoked and shows here the importance of the psalter. The poem also reveals the acrostic "Heriveus scripsit me S[an]c[t]o Bertino" ("Heriveus wrote me at Saint-Bertin"). Therefore, the monk Dodolinus prepared the literary pieces, the monk Heriveus copied them and Odbert made the decoration. Odbert surely supervised the choice of scenes and their distribution in the manuscript, and he produced the three full-page paintings framing the preface texts and the thirty-two historiated initials punctuating the text of the Psalms. The illuminations are mostly made in gold and silver in a fine and sophisticated design. The refined decoration and the skillfully elaborated assembly of texts in the Odbert Psalter are explained by the fundamental importance of the psalter for the Bertinian monastic community.

Like all communities following the Benedictine rule, the monks of the Abbey of Saint-Bertin devoted a large part of their time to contemplation and meditative practices, allowing them to reach the knowledge of God in a beatific vision, guided by the Holy Spirit. These meditative practices were intended to purify the monk's soul and to bring about his spiritual progress, which would lead him to a union with God. They consisted of prayers that took place, for the most part, during the monastic services that punctuated the lives of the monks (Davril and Palazzo 2000, p. 122). Saint Benedict, in his Rule, defines the eight offices of the day with the aim of incessant prayer: Matins or Vigils, Lauds, the office of Prime, Terce, Sext, None, Vespers and Compline (Schmitz 2018). At each of these hours, the Psalms are recited. The use of the psalter during the liturgical office was quickly defined and codified by various monastic rules, and the *Rule of Saint Benedict* granted this book an essential role in liturgy and private prayer. Chapters 9 to 18 of the Benedictine Rule are devoted to the distribution of the Psalms during the Divine Office (Schmitz 2018). It is of rare precision concerning the place of the Psalms in the liturgy of the hours: Saint Benedict specifies the individual the Psalms that must be sung at each hour of the liturgical office. He recommends reciting the entire psalter each week and therefore rigorously distributes the 150 Psalms throughout the canonical hours of the week.

The psalter, a book of the Old Testament, was extracted early on from the biblical manuscript to meet the needs of the liturgy of the Hours (Palazzo 1993, p. 148) It is indeed an ideal support for the spiritual and moral instruction of the Christian, the Psalms being lyrical poems that allow the elevation of thought and the exaltation of religious feeling. The psalter is the main source of individual and common prayers for Christians; it is considered a prophetic book, a mystery containing the New Testament. Therefore, it was particularly suitable for spiritual meditation, which also played an important role in private devotion. Saint Benedict also expressly required monks to memorize the Psalms and study them through meditation. The Psalms were therefore just as much intended to be sung during the office as meditated on; similar to prayer, psalmody was of central importance in the Benedictine contemplative life.

In accordance with the role of the psalter defined by Saint Benedict in his rule, the Odbert Psalter was composed with a view to liturgical use so that the Psalms are recited during the monastic office. They are indeed distributed according to the feria (or days) of the week. Furthermore, the images that decorate the psalter represent liturgical rituals or evoke them through the representation of hosts, altars and chalices. The dedication poem in Odbert's manuscript, identifying the authors on folio 1 verso, also clearly informs us

about the role of the psalter in the life of the monastic community. The verses "*Coenobiique Sithiensis sic concio sancta/Rite Deo psallit, quorum penetralibus altus/Istud opus coeptum, Domino patrante, peregi*" ("And thus the holy company of the monastery of Saint-Omer/solemnly sings Psalms to God, the clear sound of which [is] in the sanctuary", according to a translation by Kirsten Lynn Milliard (2013, p. 129), confirm that the Psalms contained in this manuscript were indeed intended to be sung during the liturgy in the sanctuary. "De rite," meaning "according to the rite" (translated as "solemnly" by Milliard), supports the liturgical dimension of this chant. The psalter was therefore omnipresent in the life of the Bertinian monks and constituted one of the most fundamental elements of prayer and liturgy.

Enthusiasm for psalmody increased to a considerable degree during the monastic reforms of the second half of the 10th century, when it became one of the first duties of monks. In 944, the Count of Flanders Arnulf I (918–965) decided to reform Saint-Bertin and other Flemish monasteries with the help of the reformer Gerard of Brogne (Vanderputten 2013). Since we have no concrete information about the functioning of Flemish monasteries before or after these reforms, we cannot measure their real impact. However, the Bertinian establishment was also in contact with the abbeys of southern England. There are several proofs of exchanges between the Anglo-Saxon abbeys and Saint-Bertin: missions of Flemish monks in England and vice versa, as well as a correspondence between the abbot Odbert and the archbishops of Canterbury, Aethelgar and Sigeric (988–990 and 990–991). The letters have been copied in two manuscripts: London, British Library, Cotton Tiberius A. xv, fol. 145v–146v and 161v–162v, and Cotton Vespasian A. xiv, fol. 160r and 159rv (Vanderputten 2006). At the end of the 10th century, reforms had also taken place in Anglo-Saxon monasteries, where the *Regularis Concordia* (or *Regularis Concordia Anglicae Nationis Monachorum Sanctimonialiumque*) was applied from then on. This monastic rule was written around 970 in Winchester at a council convened by Aethelwold, Bishop of Winchester, and King Edgar (959–975). It enormously increased the psalmody during the office in the reformed Anglo-Saxon monasteries. These new monastic customs undoubtedly influenced those of the abbey of Saint-Bertin, which was reformed in turn. The incessant recitation of the Psalms makes it possible to raise the soul of the Christian toward God in order to lead him to a conversion, hereby purifying his soul. Psalmody therefore constituted, as we will see, one of the modalities of contemplative practice in that it allowed the mind to immerse itself in meditation on the words of God. Contemplation, an ecstatic experience in front of God, sometimes called a "vision", is not only a vision of God; it is also to bathe in his light and his Truth, which can only be known by meditating on his Word, which is contained in the Psalms.

From the end of the 10th century, the increased role of the psalter in the Benedictine liturgy was also due to the purifying power of the psalter in developing penitential contexts. The text *Dicti Sancti Augustini quae sint virtutes psalmorum* ("Words of Saint Augustine on the Virtues of the Psalms") witnesses this preoccupation, which was shared by the monks of Saint-Bertin. This text was indeed copied at the beginning of the Odbert manuscript, even before the text of the Psalms, on folio 6 verso. The *Dicti Sancti Augustini* was composed to help the penitent use the Psalms and lists the merits of psalmody. Kathleen Openshaw (1993) points out that this text appears in some Anglo-Saxon psalters of the 11th century. It evokes on several occasions the protective power of the psalter against Evil and temptations and insists on the capacity of the chanting of the Psalms to purify the soul of the Christian and elevate him to help him enter the Kingdom of Heaven: "He who loves the diligent singing of the Psalms cannot sin [. . .] and his soul will be purified" ("*Qui diligit canticu(m) psal moru(m) assidue, non potest peccatu(m) agere [. . .] (et) anima(m) sua(m) in c(o)elo purificabit.*"). In the Odbert Psalter, the purification of the soul was therefore associated with psalmody even before the beginning of the Psalms. Moreover, this purification is the condition for elevating the Christian's soul and acceding the vision of God, as stated in the Gospel of Matthew: "Blessed are the pure in heart, for they will see god" (Matthew, 5: 8). Barbara Baert points out that Matthew 5: 8 is central to the reflection on spiritual seeing (Baert 2014,

p. 89). This sentence had been commented on by theologians like, among others, Bede the Venerable (*Homily*, 11:15) or Saint Augustine in *De Genesi ad litteram*, in which he states that the beatific vision of God is indissolubly related to the state of purity of the Christian's soul (Augustine of Hippo 1866b, XII, 28: 56). In *De doctrina Christiana*, Augustine goes further by asserting "we must purify our heart to make it capable of perceiving this divine light and of becoming attached to it once it has contemplated it" (Augustine of Hippo 1866a, 10: 10). Thus, the contemplation of God is only accessible by purifying the Christian's soul, and as we have seen in the *Dicti Sancti Augustini*, this purification is possible thanks to psalmody. The meditation of the Psalms therefore elevates the soul of the Christian, purifying it for the Christian to contemplate God.

#### 4. The Image of Pentecost in the Psalter

The psalter was therefore one of the most fundamental books for liturgy and monastic life. In accordance with its important status, the is carefully crafted and has sophisticated decoration. The latter includes three full-page paintings at the beginning of the manuscript, thirty-six historiated initials and seven marginal drawings, which were added after the main realization of the manuscript. These numerous images represent verses from the Psalms or scenes from the Old and New Testaments. Most of the scenes depict, among other things, the path to Salvation and the battle between Good and Evil: Christ armed, victorious against the Devil or Death, in accordance with the status of the psalter in penance. The main decoration of the psalter consists of twenty-four initials containing episodes from the life of Christ. It is not usual to see the psalter, an Old Testament text, pictured with Christological scenes; the Odbert Psalter is the first known psalter to present such a developed cycle dedicated to Christ. In their studies on the Odbert Psalter, Victor Leroquais (1941) and Rainer Kahsnitz (1988, 2004) approached the illuminated decoration and tried to identify the different scenes. They also questioned the organization of the Christological scenes; since the latter are not arranged in narrative order, Rainer Kahsnitz concluded that their arrangement is random (Kahsnitz 2004, p. 155) without examining the links of the images with the texts with which they are associated. However, an in-depth study sheds light on the reasons that presided over the location of the scenes. For example, the size of the decorated initials differs according to three different modules, and the most important scenes in Christian dogma (Pentecost, the Nativity and the Crucifixion) take place in the most imposing initials, revealing a hierarchy in the distribution of scenes. Some Christological episodes whose iconography evokes liturgy and rituals, such as the censing of the altar in the scene of the Annunciation to Zacharias, are next to the Psalms sung during these rituals (the Annunciation to Zacharias appears in the initial of psalm 41, and psalm 42 is chanted just before the censing of the altar). Finally, links between the text and the images also guide the distribution of scenes. These links are based on the Christological interpretation of the Psalms. Indeed, the commentary in the margins, inspired by Saint Augustine's psalm commentary, delivers a neotestamental interpretation of these poems of the Old Testament. The Christological cycle can therefore be considered a real visual exegesis of the psalmic text, offering a commentary in image. Thanks to these observations, it is possible to provide an explanation for the surprising choice to represent the Pentecost scene at the beginning of the manuscript.

The Pentecost scene on folio 11r shows a particular iconography. The folio presents an architectural border made of columns and a semicircular arch (Figure 1). In place of what should be the bases and capitals of the columns are, respectively, trilobes and quatrefoils containing scenes from the life of David (*I Samuel*, 1: 34–37, 48–50,51, 54–58). The space delimited by the architectural motif is painted in purple. It contains the text of the first psalm, called "*Beatus vir*", copied in gold letters. The initial "B" of the text, which is of a large size and located in the center of the composition, is made of gold and silver (Figure 2). In the upper body of the B (Figure 3), Christ, dressed in gold and silver, is enthroned and points with one hand to the Book, his Word contained in the New Testament. He is framed by curtains drawn with a white line on an ash-blue background. The initial visually

introduces Christ for the first time in the manuscript as an enthroned king. In the lower part of the initial (Figure 4), the twelve apostles are represented; the three in the foreground rest their chins on their hands in a gesture of meditation. Above them, a dove head toward them in a downward movement. This scene has been interpreted as that of Pentecost, recounted in the Acts of the Apostles: "When the day of Pentecost came, they were all together in one place. Suddenly a sound like the blowing of a violent wind came from heaven and filled the whole house where they were sitting. They saw what seemed to be tongues of fire that separated and came to rest on each of them. All of them were filled with the Holy Spirit and began to speak in other tongues as the Spirit enabled them" (*Acts of the Apostles*, 12: 1–4). This is the episode in which, after ascending to heaven during his Ascension, Christ unveils the New Law to the apostles and asks them to spread his Word throughout the world.

This episode comes after the Incarnation, the sacrifice of Christ and his Ascension, and just before the mission of the apostles to spread the Gospel. It therefore generally appears at the end of the Christological cycles, often constituting the last scene of the narrative in images. It is therefore surprising to find this scene as the opening of the psalter and of the Christological cycle if we look at the chronology. However, the text of the first psalm can justify the location of this episode: "Blessed is the man who has not let himself go to the counsel of the ungodly [. . .]/But who has his pleasure in the law of the Lord, and who meditate on it day and night" ("*Beatus vir qui non abiit in consilio impiorum et in via peccatorum non stetit [. . .]/Sed in lege Domini voluntas eius et in lege eius meditabitur die ac nocte*"). Also, the Law of God was fully revealed at Pentecost. Christ brandishes this New Law in the form of the Book. The apostles represented in the foreground, below Christ, are in an attitude of meditation; they can therefore be fully assimilated to this blessed man who meditates on the Law of God. The monks could then easily identify with the apostles, as they were considered the heirs of the vita apostolica. The Pentecost image could have been placed at the beginning of the text of the Odbert Psalter in order to advise the monastic reader from the outset to model himself on the apostles and, like them, to meditate on the Law of God contained in the Psalms.

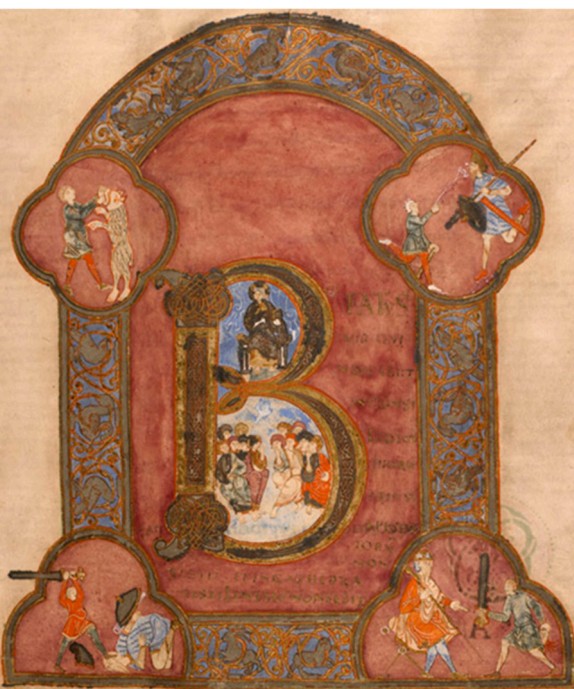

**Figure 1.** Boulogne-sur-Mer, Bibliothèque municipale, ms. 20, fol. 11 recto. © 2013 Institut de recherche et d'histoire des textes.

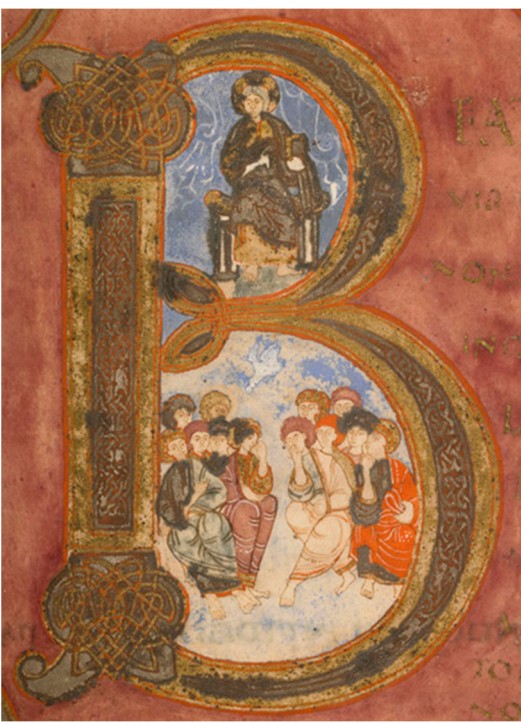

**Figure 2.** Boulogne-sur-Mer, Bibliothèque municipale, ms. 20, fol. 11 recto. © 2013 Institut de recherche et d'histoire des textes.

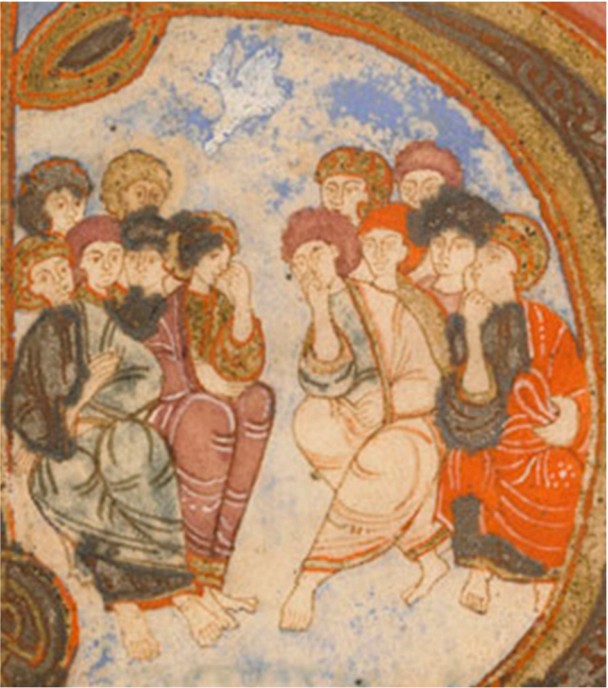

**Figure 3.** Boulogne-sur-Mer, Bibliothèque municipale, ms. 20, fol. 11 recto. © 2013 Institut de recherche et d'histoire des textes.

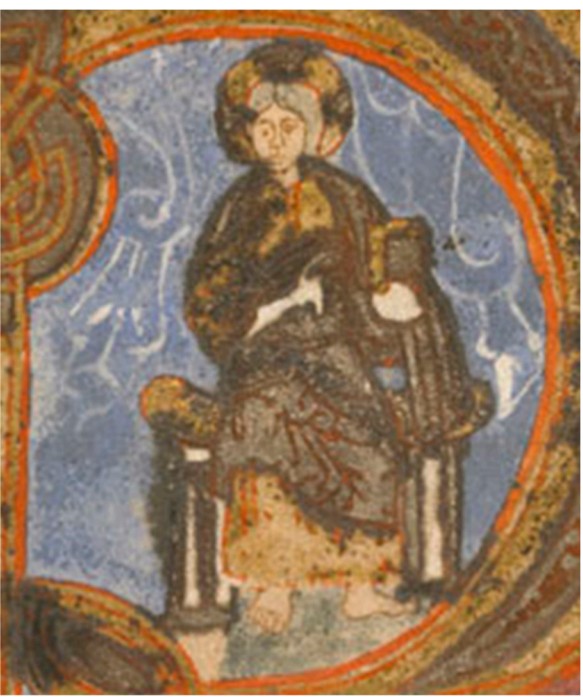

**Figure 4.** Boulogne-sur-Mer, Bibliothèque municipale, ms. 20, fol. 11 recto. © 2013 Institut de recherche et d'histoire des textes.

However, Pentecost is represented here in an unusual way. Although the dove of the Holy Spirit descending on the apostles allows us to idenfty this scene, the tongues of fire that are mentioned in the text of the Act of the Apostles are omitted here. They are figured in most of the scenes of Pentecost, for example, in the *Benedictionary of Aethelwold* (London, British Library, Add MS 49598, folio 67v, realized in Winchester between 963 and 984) or a sacramentary, a contemporary of the psalter, which is now in Rouen (Rouen, Bibliothèque municipale, ms. 0274 (Y. 006), folio 84v, realized in Canterbury circa 1020) (Figure 5). These two illuminations present a rather similar composition in which the dove of the Holy Spirit, in a mandorla carried by two angels, flies toward the twelve apostles. Flames, the tongues of fire, descend from the mandorla on the heads of the apostles. The figure of Christ painted in the ornamented letter of the psalter is also very rare: at the end of the 10th century, in images, the presence of God is manifested only by the dove of the Holy Spirit, as in the above-mentioned manuscripts, or by the Hand of God. The illumination of the psalter deviates from the New Testament narrative in which only the coming of the Holy Spirit upon the apostles is mentioned. In addition to the case of the Odbert manuscript, in the Drogon Sacramentary (Paris, Bibliothèque Nationale de France, MS lat. 9428, realized in Metz between 845 and 855, folio 78r), the dove of the Holy Spirit in a mandorla descends on the apostolic college while golden rays, constituting the tongues of fire, reach the head of each apostle (Figure 6). Christ is represented on a cloud and places his hand on the dove of the Holy Spirit. The attention here is focused on the Holy Spirit, unlike the Odbert Psalter in which the entire upper body of the initial B is devoted to Christ. The importance given to him can, however, be justified, thanks to the important role of the psalter in the contemplative life.

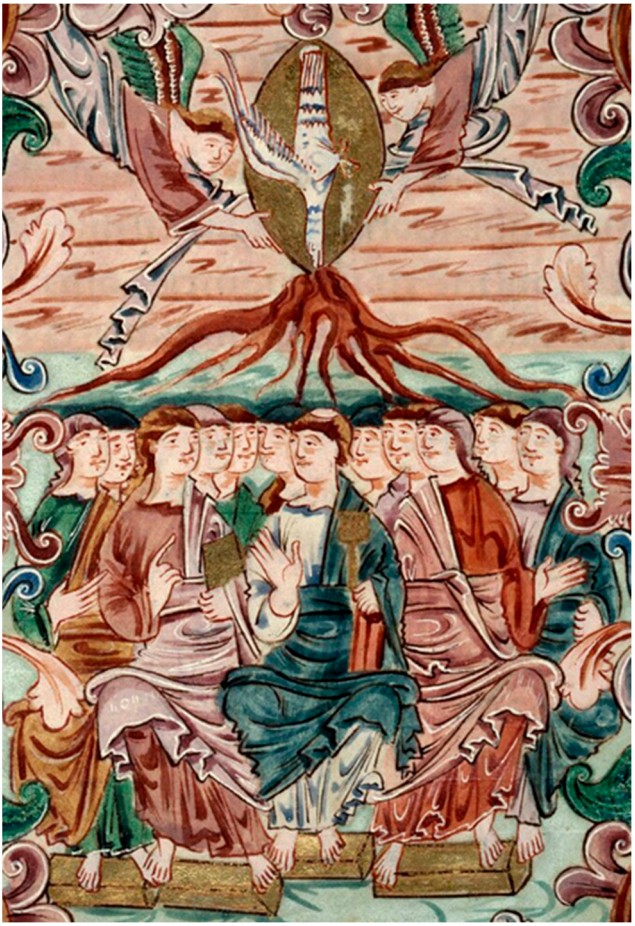

**Figure 5.** Rouen, Bibliothèque municipale, ms. 274 (Y. 006), folio 84 verso. © 2013 Institut de recherche et d'histoire des textes.

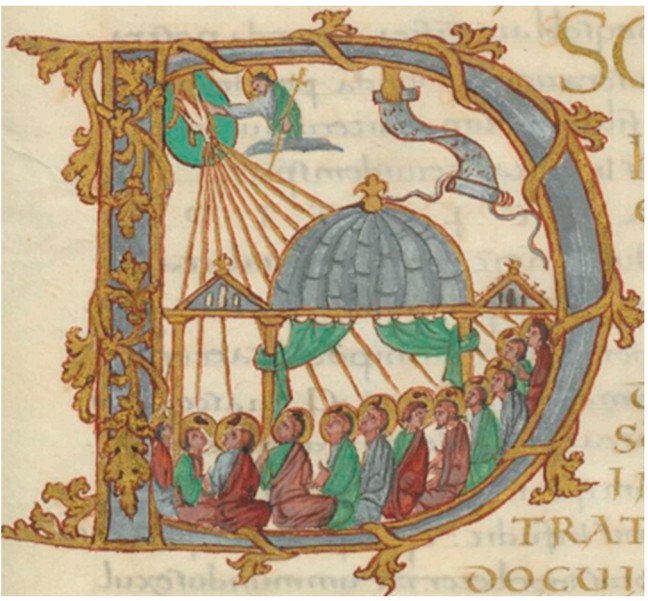

**Figure 6.** Paris, Bibliothèque Nationale de France, MS lat. 9428, folio 78 recto. © Source gallica.bnf.fr/Bibliothèque nationale de France.

Christ is not mentioned in the verses relating the episode of Pentecost. In the image, even though he is figured, he does not seem to be part of the narration. He stands in a

well-defined space separate from that of the apostles: the upper body of the B is therefore a celestial and sacred space to which only the divine has access. Figured as a king enthroned, statically, this image is a true portrait of Christ, enthroned for eternity. It would be an image, or imago, of Christ that represents him in a timeless way. This image is therefore not part of a narrative whole. Moreover, the apostles look straight ahead and not toward Christ: this means that the latter is not present in the scene and that he does not appear to them in "corporeal" vision. If the creator of the image chose to represent Christ when he does not manifest himself corporally to the apostles, it is perhaps because he appears to them in a "spiritual" vision or "spiritual seeing", in the words of Barbara Baert (Baert 2014). In her article, she interprets the iconography of a Pentecost scene in the *Codex Egberti* thanks to the different levels of vision defined by Saint Augustine (Baert 2014, 2016). The works of this Church Father can also be consulted to interpret the image of Pentecost in the Odbert Psalter; indeed, his thought is present in the psalter with his commentary on the Psalms, providing a Christian interpretation of the Psalms. Moreover, his works were copied in a sustained manner in the scriptorium of Saint-Bertin. For example, the inventory *Brevis annotatio librorum sancti Bertini*, whose main witnesses are Besançon, Bibliothèque municipale, ms. 1106, and Académie 8, copied at the end of the 11th century, mentions twenty-six books of Augustine of Hippo kept in the monastic library of Saint-Bertin, making him the most represented author in this Bertinian inventory. The Bertinian monks may therefore have been familiar with the Augustinian theories about the visions of man and their integration into a means of accessing the contemplation of God.

## 5. The Three Modes of Vision of Saint Augustine

Saint Augustine distinguishes several elements in the human person: the body and the soul, which is itself formed by the spiritus and the mens. The spiritus is what vivifies the body and is also possessed by animals. The mens is the part of the soul possessed by man alone which allows for access to intelligible truths, bringing man closer to God. Saint Augustine deduces from his ternary vision of man a distinction between three types of vision: corporeal vision, which is formed in the soul from the eyes of the body which observe a material image, a physical object which strikes our gaze. Then comes spiritual vision, which concerns the creation of mental images in the semblance of the bodies but without seeing anything with the eyes of the body. It allows us to see things in their absence, whether they are things that we have actually observed and which memory reproduces or whether they are the fruit of our imagination (Augustine of Hippo 1866b, XII, 20: 22). Finally, intellectual vision, or the vision of the heart, constitutes the last level and concerns the ideas to which there does not correspond any image which represents them exactly, such as God and his Truth. Whereas the first two visions are called corporeal and spiritual because they depend, respectively, on the perception of the senses of the body and the spirit, the third is called intellectual because the image and the Knowledge of God are reflected in the intellect (Augustine of Hippo 1866b, XII, 7: 18).

It is at this last level that Augustine places the "*visio Dei*", the intellectual vision that allows for the vision of God. In the first two types of sight, God can manifest himself through corporeal or imaginative visions, but he only reveals himself indirectly, in an appearance which represents him (as we have seen, the image is indeed based on the principle of resemblance between the representation and the intelligible subject or prototype). He reveals himself in a more immediate and perfect way in the intellectual vision. The word "vision" is not used here in the literal sense of the term: intellectual vision is the understanding of the divine essence. Augustine defines intellectual "images" as ideas independent of matter which we conceive and define only thanks to reason: God, the soul, reason, virtue, prudence and justice, for example. This vision, says Augustine, is an act of the intellect freed from all material perception: "We actually reach these ideas without distinguishing either design or color, without perceiving either sound, smell or taste, finally without being warned by the touch whether there is a cold or warm, hard or soft, rough or polished surface; we are guided by another light, another brightness, another sight

more infallible than the sensations and much higher" (Augustine of Hippo 1866b, XII, 3: 6). Intellectual vision is indeed, for Augustine, the highest vision. Removed from the influence of the senses and the imagination, the soul no longer sees anything but purely intelligible truths. It is only at this last level that man can contemplate God in his clarity, no longer seeing through the clouds of a sensible vision: "we see him face to face and without a veil, such as the human soul can to understand, such as his grace reveals to those whom he deems worthy of participating more or less intimately in the conversation where he speaks directly to intelligence" (Augustine of Hippo 1866b, XII, 26: 54).

To access intellectual vision, Augustine indicates that man needs the intellectual power to abstract things from their material substrate for the benefit of the sole vision of the heart, an act of the intellect devoid of any bodily semblance and freed from all sensible impression. Vision is therefore a cognitive, difficult and hierarchical process that aims to reach a state of spiritual bliss that allows man to see (in the sense of understanding) what is beyond the visible and to reach an understanding of the divine essence—in other words, to contemplate God. This contemplation is accessible through meditation and reflection on the Word of God and depends on the Christian's ability to disregard the material and to concentrate his mind during prayer in order to direct all his efforts toward a supreme goal: the vision of God.

In the image of Pentecost, the apostles are in a meditative attitude. The abbot may have painted in the upper body of the letter the inner vision which the disciples attained as they were filled with the Holy Spirit, a contemplation guided by the action of the Holy Spirit. It was through their meditation on the Psalms that the apostles raised themselves intellectually and arrived at this vision of God. Odbert therefore painted what each monk should aspire to and what the monastic community should achieve. As we have already pointed out, monks can easily identify with the apostles since they are considered their heirs. Thus, by meditating on the law of God contained in the Psalms that he is about to read during the office or in the context of the exercise of private prayer, a monk will also be able to access this intellectual vision and contemplate God. In the Odbert Psalter, the letter contains the image; the reading of the Psalms is therefore fully associated with the vision of the image, which stages the elevation of the soul toward the vision of God. Thus, the image, in the same way as prayer and meditation on the Psalms, also has an important role to play in the modalities of intellectual vision.

In fact, Augustine's three levels of interpretation were transferred to the domain of the image. In *De vero religione* (written circa 390) and then in book XII of *De Genesi ad litteram*, the bishop of Hippo already distinguishes the three types of visions mentioned above and hierarchizes them. Augustine maintains that existing things and images that act as "visible words" are signs given by God, the Creator. As a result, they show divine truth. This "is not visible to our eyes [by corporeal vision], and no more to the eyes which make us see the images imprinted in the soul by the eyes of the body [by spiritual vision]" (Augustine of Hippo 1864, 39: 73) but is due to intellectual vision. While the eyes of the body perceive these images physically, the inner eyes implement the "spiritual gaze" to access divine truth. Thus, according to Augustine, a material image can be a starting point or even a "springboard" for an ascent toward the intelligible and the process of visionary practice. "It is on the place where you have fallen that you have to lean to get up; it is precisely on the carnal forms that captivate us that we will lean to know those that the flesh does not manifest" (Augustine of Hippo 1864, 24: 45). The itinerary toward the ultimate vision is therefore cumulative, with each of the visions defined by Saint Augustine constituting a stage: the corporeal vision of the image will allow access to the spiritual vision, having imprinted the image in the mind, which will allow the soul finally to arrive at the intellectual vision to contemplate God.

The image of Pentecost in the Odbert Psalter can therefore be considered the materialization of a superior vision. This interpretation could explain its particular iconography, in particular the highlighting of the figure of Christ. The text of the psalm or the commentary may explain some of the iconographic choices of the Pentecost scene, but this scene goes

beyond what is expressed in the text. The Word is finally unveiled and revealed in the image, where, as Barbara Baert states, different levels of interpretation are entangled (Baert 2014, p. 89; 2016). Thus, the representation of Christ in the upper body of the initial represents the intellectual vision that the apostles have and promises this contemplation if the reader follows the example of the apostles in meditation; it also materializes an ideal to be achieved in the contemplative life of the monk. Moreover, this image also makes it possible to contemplate by participating in the means of accessing the vision. The viewer, upon opening the book of Psalms, will see the image of Christ and will have a bodily vision of Christ. This material support, through its character of imitation, will allow him a mental projection in the viewer's spirit, leading to a spiritual vision; then, by dint of meditation, he will be able to access the vision of the heart and thus truly contemplate God. It is also the association of the image and the text which will allow for this vision: the Psalms, a text of prayer and meditation par excellence, makes it possible to elevate the soul of the reader while the image stimulates the religious feelings of the spectator and implements the "spiritual" gaze. Moreover, Christ points to the Book with his finger: he surely presents it as a tool for accessing this contemplative vision. The image of Christ in the scene of Pentecost can therefore constitute a means of ascent from bodily vision to intellectual vision, enabling a passage from the material object to the spiritual. The mental translation of the viewer starts from the physical vision of this enthroned Christ, moving toward the same image but of a different nature which results from the vision of the heart.

## 6. Conclusions

The spirituality and liturgy of the monastic community of Saint Bertin, oriented toward the contemplative life, is based in part on the chanting of the Psalms. The psalter constituted a large part of the monks' prayers from the establishment of the Benedictine rule in the sixth century, but the reforms of the tenth century reaffirmed and surely reinforced the major place of this work in monastic spirituality. Also, the image, in the 10th century, responded to a need for piety and the spiritualization of the material (Baschet 2016, p. 74). Medieval practices surrounding images permit us to understand the singular iconography of the Pentecost scene and the presence of Christ in it: its function is to stimulate the religious feelings of the spectator by constituting a means of ascent from the material to the spiritual. The study of the Odbert Psalter makes it possible to better understand not only medieval spirituality and contemplative practices but also that images had an essential role there. Text and image participate jointly in the implementation of contemplation: the reading of the psalter allows the monk to meditate on the Word of God and purifies his heart in order to prepare it for beatific vision; then, the image allows the viewer, through the material vision, to project beyond it in order to reach the intellectual vision of God. Through this association of the text of meditation and the image–springboard, the Odbert Psalter, as a whole, becomes a major tool of devotion for the Christian. By constituting a support for the contemplation of God, the psalter is the perfect mediator of the power of the intellect.

**Funding:** This research received no external funding.

**Institutional Review Board Statement:** Not applicable.

**Informed Consent Statement:** Not applicable.

**Data Availability Statement:** Not applicable.

**Conflicts of Interest:** The author declares no conflict of interest.

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
