# Peer review of "The Odbert Psalter (Boulogne-sur-Mer, BM, ms. 20); or, the Image as a Medium for Contemplative Practice"

_religions, doi:10.3390/rel14091213_

Round 1

Reviewer 1 Report

At the outset, the author states the unusual layout of the Odbert Psalter in terms of starting with the psalms used at Pentecost (the place of which is normally at the end of psalters) and the appearance of Christ in the pictorial scenes accompanying the psalm texts (according to the biblical texts he dwells no longer on earth at Pentecost). She/he sets out to explain this unusualness by tracing the practices surrounding the images. “Enthusiasm for psalmody increased to a considerabe degree during the monastic reforms of the second half of the 10th century, when it became one of the first duties of monks. […] Psalmody constitutes […] one of the modalities of contemplative practice in that it allows the mind to immerse itself in meditation on the words of God.” (L. 105-107, 117-119).

The author explains the surprising appearance of the Pentecost scenes at the beginning at the psalter with the psalm texts accompanying it, which refer to the law of the Lord (that has been unveiled to the apostles at Pentecost): “The Pentecost could have been places at the beginning of the text of the Odbert Psalter in order to advise the monastic reader from the outset to model himself on the apostles and, like them, to meditate on the Law of God contained in the psalms.” (L. 213-216). The second unusual item, the appearance of Christ in the scene, is explained by his depiction being in a separate space: “The upper body of B is therefore a celestial and sacred space to which only the divine has access. […] If the creator of the image chose to represent Christ when he does not manifest himself corporaly to the apostles, it is perhaps because he appears to them in a “spiritual” vision.” (L. 261f., 267-269). The background to these different modes of vision is supposed to lie in the definition of visions of different nature by St. Augustine. The way of depiction is said to reflect medieval pictorial practices which demanded veneration for the picture as a representation (and an object of mediation) of the prototype (God or saints).

The argumentation of the author is clear and easy to follow. His/her conclusions rest on a sound basis of scientific literature and on a deep understanding of medieval liturgical practice. The publication of the article in its present form is recommended.

Spelling mistakes:

L. 82: individual and commo [sic!] prayers of Christians

L. 376: Medieval images were in fact based on the principle of conjoining [the] material and [the] spiritual (again in L. 393)

Author Response

Thank you very much for your reviews. The spelling mistakes had been corrected and the English reworked and improved.

Reviewer 2 Report

It is a fascinating article pointing out a significant phenomenon and nicely connecting the visual analysis of the depicted miniatures with the historic settings of 10the century Benedictine reform spirituality and of there apects of contemplation, as conceived in the Latin Augustinian tradition.

The paper is very well written and I have found no unclarities or typos.

As for the content I have only one remrak\request. It could have been most helpful and perhaps also significant if the author could provide us with a brief description of the other illuminations in this manuscript. Mostly interesting in my point of view would be to know whether any of the other illuminations entails some similar hints for contemplative practices. Both positive and negative responses might bear significance to the discussion.  

Author Response

Thank you very much for your reviews. Here is my reply to your report:

Remark: It could have been most helpful and perhaps also significant if the author could provide us with a brief description of the other illuminations in this manuscript. Mostly interesting would be to know whether any of the other illuminations entails some similar hints for contemplative practices.

Response: I have made some modifications to better describe the other images of the Psalter,but it would be difficult to do this more precisely without devoting a very long paragraph to them, because there are 46 of them and of very different subjects. In previous researches, I have pointed out that most images figured battle of Good and Evil, which seems to be the main "theme" of the decoration, so I mentioned this point and referred to a bibliography. At this point I did not find similar hints for contemplative practice in the other images but I will not affirm that there are none either.

Reviewer 3 Report

This article wonderfully highlights the turn to contemplative monastic life and its reflection in the manuscript tradition, particularly in the new use of the scene of Pentecost in the Odbert Psalter. The research on this manuscript appears to be original and the writing is easy to understand.

I have some minor editorial comments to follow, but first I wanted to express a few questions. I was never very clear on the corpus of Pentecost images in 10th-century manuscripts (ones without rays of light and the dove). Is this really the first time we see a contemplative version of the scene in a manuscript? That point should come out more strongly if true. If it is not the case, then what makes this image unique?

I also wondered if St. Augustine had a special role at Saint-Bertin, as in - it is not an Augustinian monastery, right? Why is St. Augustine so important in the 10th century at this location? Everyone says that his vision theory is famous - is that merely the case here, or were other 10th-century sources at Saint-Bertin quoting Augustine and making his vision theory freshly relevant at the monastery? Along those line a footnote about the reception of Augustine might be worthwhile. I'm thinking about sources like Jesse Keskiaho, Dreams and Visions Ch 4.

While I find the argument about the 3 levels of vision convincing, I wondered if more might be said about its intellectual framework. For example, what was the significance of Christian Neoplatonism in the 10th century? Were they interested in Ambrose's vision theory? Were they looking at other sources? Here I worry that we don't have as deep a discussion about corporeal sight to spiritual vision that we could have. I'm thinking about B. Baert's other article (not listed in the bib here) in Convivium 2014. In fact, her arguments are intellectually close this this current article and it would be good to distinguish her framework from the one here or at least acknowledge the intellectual debt to her article, which clearly informed the thinking. Baert refers to Bede and others beginning on p. 108...

As far as editing goes, the whole would benefit from a careful copyedit, esp the footnotes where I see several minor mistakes.

In the abstract, Sainte Augustine doesn't need an "e"

"The" is used as the first word of several sentences in the paragraph beginning at line 34

Line 48 - this section needs a topic sentence that explains the purpose of the paragraph and the section in terms of the overall thesis.

line 56 - Odbert therefore surely seems awkward to me and also forced

line 82 - commo?

lines 84-87 - tenses

line 107 - Arnulf?

line 108 - UC Flemish

Lines 114-115 - this sentence is unclear. Restored?

Lines 117-120 - we need a source for these ideas, esp a 10th-century source?

line 126 - lc evil?

line 130 "quoting Matthew "Happy" - awkward. 

Line 138 and following - What is this section about? Signpost

Line 143 - this sentence is unclear

Line 157 - which gloss? unclear

Line 158 - awkward sentence. 

Line 180 - need a topic sentence here. What is this paragraph about? Architectural frames?

Line 203 - UC Mission of the Apostles?

Line 205 - I'm not surprised to find Pentecost not included in a Christological cycle or at the end of it. The sentence needs refining. 

Line 249 - chirophany? Maybe define this if it is a technical term? I never heard of it.

Line 251 - sentence beginning with "Besides" is awkward.

Line 259 - another awkward sentence.

Line 317 - "It"??

Line 365 - need " instead of >>

Author Response

Thank you very much for your reviews. Here are the response to your report :

Point 1 : I was never very clear on the corpus of Pentecost images in 10th-century manuscripts (ones without rays of light and the dove). Is this really the first time we see a contemplative version of the scene in a manuscript? That point should come out more strongly if true. If it is not the case, then what makes this image unique?

Response to point 1 : There are rare image of Pentecost before 10th and even 11th century, and this is the only one known both with Christ and without rays of light. I better emphasized that this image is quite unique due to its iconography (not because of the contemplative version of the Pentecost) and that the point is to explain the strange iconography by the contemplatives practices around images.

Point 2 : I also wondered if St. Augustine had a special role at Saint-Bertin, as in - it is not an Augustinian monastery, right? Why is St. Augustine so important in the 10th century at this location? Everyone says that his vision theory is famous - is that merely the case here, or were other 10th-century sources at Saint-Bertin quoting Augustine and making his vision theory freshly relevant at the monastery? Along those line a footnote about the reception of Augustine might be worthwhile.

Response to point 2 : Augustine has not a particuliar role in Saint-Bertin, he was simply a author of great importance in all christian Western Europe and in Benedictine monasteries. I made a note about the fact that the monks were surely familiar with his ideas because he was perhaps the most copied author at the monastery (l.359-369), and it was interesting to see Augustine's theory because the psalter presents several texts of him in the manuscript.

Point 3 : While I find the argument about the 3 levels of vision convincing, I wondered if more might be said about its intellectual framework. For example, what was the significance of Christian Neoplatonism in the 10th century? Were they interested in Ambrose's vision theory? Were they looking at other sources? Here I worry that we don't have as deep a discussion about corporeal sight to spiritual vision that we could have. I'm thinking about B. Baert's other article (not listed in the bib here) in Convivium 2014. In fact, her arguments are intellectually close this this current article and it would be good to distinguish her framework from the one here or at least acknowledge the intellectual debt to her article, which clearly informed the thinking. Baert refers to Bede and others beginning on p. 108...

Response to point 3 : This article figures in my bibliography in its French version: 2016. “La Pentecôte dans le Codex Egberti et le Bénedictional de Robert de Jumièges”. In Les cinq sens au Moyen Âge. Edited by Éric Palazzo. Paris: Éditions du Cerf. Pp. 523-528, that seemed more suitable to me because of the language but also the more recent date. I can indicate the Convivium reference to avoid misunderstandings, because this is verbatim the same text. Given the format of the article I preferred to focus on image and vision rather than a deep discussion about monastic spirituality. About theories of vision, Baert cites in her articles Bede and Ambrose but not about vision - or only for a Bede's commentary on Matthew 5:8 for which I cite Saint Augustine. Both authors were indeed copied at Saint-Bertin but there is no way to better determine the way monks were influenced, and Sainte Augustine remains more suitable as I said in my precedent response, because the psalter contains texts from him.